# *Libro del Edificio Electrónico* (LdE-e): Advancing towards a Comprehensive Tool for the Management and Renovation of Multifamily Buildings in Spain

**Paúl Espinoza-Zambrano** [1,*] **, Carlos Marmolejo-Duarte** [1,*] **and Alejandra García-Hooghuis** [2]

1 Centre for Land Policy and Valuations (CPSV), Department of Architectural Technology (TA), Architecture School of Barcelona (ETSAB), Universitat Politècnica de Catalunya, 08034 Barcelona, Spain
2 Account Manager at Minsait (Indra), 28108 Madrid, Spain
* Correspondence: paul.espinoza@upc.edu (P.E.-Z.); carlos.marmolejo@upc.edu (C.M.-D.)

**Abstract:** In 2018, the Energy Performance of Buildings Directive (EPBD) introduced Building Renovation Passports (BRP) to enable buildings to scale energy performance through a Scheduled Renovations Roadmap (SRR). The Digital Building Logbook (DBL) was introduced in 2021 as a repository of relevant building data to facilitate informed decision-making and information sharing among stakeholders. In Spain, both tools (DBL + SRR) have been studied separately in an incipient way. However, the interconnection of data in the tools come from the same document base. Evidence suggests that when building information is used in isolation, its value is diluted without stakeholder awareness. In this paper, we move towards unifying both tools (DBL+SRR) in a single comprehensive tool called the *Libro del Edificio Electrónico* (LdE-e), with the aim of producing a single building database to drive multi-building renovations. For this purpose, the data fields of existing building information, assessment and management tools in Spain were studied in detail and reinterpreted in a new interconnected data structure. To evaluate the formulation of the LdE-e proposal, 11 semi-structured interviews were held with a panel of 13 experts specialized in real estate, building and energy efficiency. After these sessions, the LdE-e was reformulated, and the model was strengthened by analyzing vectors such as governance, management, usability, data flows, stakeholders and the impact of including new technologies such as BIM and blockchain. The results suggest that the LdE-e would improve control of the operation and maintenance of (new or existing) buildings, the programming of renovation actions based on deficits identified in technical inspections, and other aspects.

**Keywords:** BRP; DBL; renovation wave; multifamily housing





## 1. Introduction

Efforts to transition to a decarbonized global economy by 2050 are currently insufficient. The 2022 Conference of the Parties (COP27) concluded with new agreements to offset carbon emissions, but the slow progress has been branded as disappointing and a collective failure [1]. We have achieved less than half the progress we should have made on energy decarbonization at this stage [2,3].

The construction sector has a leading role to play in the fight against the greenhouse effect. In Europe, buildings consume 40% of final energy and emit 36% of greenhouse gases (GHG). In Spain, the situation is similar: buildings consume 30.1% of energy and emit 25.1% of GHG [4]. This reflects the existence of obsolete, energy-inefficient building stock.

The European Commission (EC) has proposed reducing GHG emissions by at least 55% by 2030 (compared to 1990 emission levels), as a prudent route to economic decarbonization by 2050 [5]. To this end, the construction sector has become important in strategies such as the European Green Deal, the Renewal Wave and the Next Generation EU temporary recovery instrument, which plays a crucial role in European economic recovery after the

COVID-19 pandemic. The idea is that buildings can be renovated intensively by local labor in the construction industry.

Proper building information management is an important element for the success of these strategies, since to communicate conventional or energy efficiency aspects of buildings, stakeholders must be convinced that what is presented as "efficient" really is so. For this purpose, the European Union (EU) has envisaged the introduction of two instruments: the Building Renovation Passport (BRP) (as a tool to enable buildings to increase their energy performance through a Scheduled Renovations Roadmap, SRR) through the Energy Performance of Buildings Directive (EPBD) (EU) 2018/844 [6]; and the Digital Building Logbook (DBL) (which is presented as a repository of relevant building data, including energy data, to facilitate informed decision-making and data sharing among stakeholders) through the recent Renovation Wave strategy [7]. Although they were presented in different years (BPR in 2018 and DBL in 2020), the relationship between the instruments is elementary, since the DBL constitutes the documentary basis of conventional and energy-related data of the BRP that is required to build the SRR.

In Spanish residential buildings, the two instruments have been studied separately in an incipient way [8–10], even though the interconnection of data in the tools comes from the same document base. Isolated study, development and future implementation of the instruments might lead to difficulties in the medium and long term, as a lack of coordination among stakeholders may diminish the value of the information, undermine investor confidence in sustainable building and further delay the attainment of climate neutrality by 2050. This issue is important in Spain, since the late implementation of Energy Performance Certificates (EPCs) and their scarce initial regulation means that the instrument has not convinced homeowners, and its impact on the decision to buy efficient homes has been limited [11,12]. Furthermore, the consensus of some experts in the Spanish real estate industry is that EPCs have not aroused the interest of demand and, consequently, of supply in efficient dwellings [13]. Therefore, the implementation of new instruments, such as DBL and BRP, must be handled wisely.

These vortices or information gaps are systemic conditions resulting from information asymmetry. They prevent users from being informed about the benefits of energy efficiency [14,15], waste time and money in the construction process [16], and lead to poor success of public policies that promote energy efficiency renovation [17].

The draft recast of the EPBD foresees that, by 2024 at the latest, Member States will have to make the BRP available to owners (on a voluntary basis). Consequently, it is vital to study this instrument. The EPBD draft introduces Minimum Energy Performance Standards (MEPS), which provide for mandatory phased retrofitting of the most inefficient building stock between 2027 and 2033 (starting, as usual, with public and non-residential stock, and continuing with residential stock).

In anticipation of the full transposition of Directive (EU) 2018/844 in Spain and the subsequent publication of its recast in 2024, this paper aims to advance the proposal of a comprehensive tool for Spain that incorporates conventional and energy information management of buildings (DBL) and an SRR (BRP). This holistic, cloud-based, blockchain-supported tool has been dubbed the E-Building Logbook (LdE-e, the Spanish acronym of *Libro del Edificio Electrónico*) since, as described below, the aim is to replace existing non-dynamic instruments but keep a familiar name that should improve its presentation to owners/users and stakeholders.

To achieve this objective, the following steps have been followed. First, Section 2 highlights the problem of information asymmetry and the formation of information gaps, which are generated when information is used in isolation. Then, a novel approach to the integration of the DBL + SRR model is analyzed to achieve the discussed LdE-e structure. Section 3 describes the methodology used, which consists of two stages. In the first stage, the initial elements for the LdE-e proposal are analyzed, that is: (1) existing building information management instruments in Spain; (2) theoretical–instrumental and normative proposals, put forward by Spanish organizations; (3) the consideration of methodologies/technologies

that can facilitate access to information by owners and interested third parties. From this basis, the isolated configuration of existing instruments is reinterpreted, and a new unified structure of interconnected data is proposed according to the condition of the building (new or existing building). This forms the first approach to the LdE-e. The second stage presents the process of semi-structured interviews with a panel of experts in sustainable building, with which the LdE-e model is evaluated and validated. Then, Section 4 describes the results of the experts' validation, discussions, limitations and future research lines. Finally, Section 5 contains the conclusions of the research.

## 2. Background

### 2.1. Information Gaps and Information Asymmetry

An enormous amount of information is generated and used during a building's life cycle. Agents in different temporal or hierarchical stages of the life cycle use this information for the benefit of their activities and the correct functioning of specific processes. However, much of the information that is produced is "created" multiple times due to a lack of coordination between the parties. Usually, the benefits of this information are diluted without the parties being aware of it. When information is generated in isolation, an information gap arises. According to Miller et al. [18], these gaps (Figure 1) are confined to four key relationships: infrastructure, valuation, regulation and consumer protection.

**INFRASTRUCTURE GAP**

- Energy supplier

- Real estate developer

- Homebuyer

- Design/build team

**VALUATION GAP**

- Valuer

- Property agent

- Homebuyer

- Financial agent

**REGULATION GAP**

- Government

- Property agent

- Homebuyer

- Home seller

**CONSUMER PROTECTION GAP**

- Government/regulatory agency

- Property agent/construction company

- Homebuyer

- Home seller

**Figure 1.** Information gaps. Source: own elaboration, based on Miller et al. (2014).

This situation is further aggravated in the case of energy efficient buildings. Although the EPC has made progress in terms of the transparency of the energy performance of buildings, energy efficiency (EE) continues to have an insufficient impact on housing marketing [19]. A chain of harmful disinformation is created, which Cadman [20] defined as The Vicious Circle of Blame. At the center of this discourse is the information asymmetry experienced by the stakeholders of the real estate market.

Market failures arising from asymmetric information have been extensively studied in conventional cases [21,22] and in the sustainable building market [23,24].

Information asymmetry is a social rather than an economic cost. Correcting it may balance interest in energy-efficient buildings. The specialized literature suggests that the creation of a shared, dynamic database of building life cycle information might be one of the first steps to overcoming the disinformation barrier, especially if it is structured using segmented criteria such as economic, technical or environmental benefits [25,26]. To systematize information on a building's traceability, a better understanding is needed of existing building data management tools, information production processes and stakeholders' information requirements during the building's life cycle. An effective information exchange would lead to a win–win situation for the demand and supply sides of the building sector and real estate in general.

*2.2. DBL + SRR—A Pipe Dream?*

In Europe, the use of BRP has been discussed for decades. Several Member States have operational models with varying functionalities depending on their real estate market and their population's degree of awareness [10]. A change of perspective is required in the approach to BRP in Spain since 71.8% of the housing stock consists of multifamily homes [27]. This condition means that the energy renovation process will require greater coordination, awareness and capacity for agreement on behalf of the owners (in Spain, the ownership of multifamily buildings is based on a kind of condominium tenure) [28]. Similarly, the correct management of building information through a DBL is relevant because the logbook must be disaggregated for each of the housing units that make up the building. The creation of the SRR involves a leap in complexity since the renovation stages must integrate different rehabilitation scenarios, according to the characteristics of each dwelling.

Far from being a handicap for the integration of DBL + SRR, the complexity of multi-ownership of buildings is an opportunity to condense building information into a single database and decrease the information gaps that can emerge if data are generated in isolation. Similarly, an overview of building renovation stages (with integrated SRR) might make owners confident about the benefits of energy renovation and provide an overview of the building stock for governments and other stakeholders. The creation of a single tool (DBL + SRR) that is cloud-based and integrated with new technologies could advance in this direction. This hypothesis, which from a theoretical perspective could be considered a pipe dream, is tested using the methodology presented in Section 3.

**3. Methodology**

The methodology proposed for this research has been organized into two phases. In the first empirical phase, existing building information management tools in Spain are identified and categorized according to the inputs they provide (Section 3.1). Similarly, the inputs that can be provided by BIM methodology and blockchain technology are analyzed. Based on these inputs, an embryonic structure of the LdE-e proposal is created, whose model is reformulated and validated through semi-structured interviews conducted with a panel of experts (Section 3.2). The second phase is of a propositional nature, in which the results of the LdE-e proposal validation are presented, based on the opinions of the panel of experts. In this phase, the data structure, data flow, governance, management, and interaction of the database with the SRR are proposed (Section 4).

*3.1. Initial Elements for the LdE-e Proposal*

The proposal for the structure of LdE-e is based on the study of existing building information management tools, theoretical and instrumental proposals and regulations posted by Spanish organizations and the consideration of methodologies/technologies that can facilitate access to information by owners or stakeholders. Therefore, the initial elements that give rise to the LdE-e (Figure 2), depending on their data contribution, are the following.

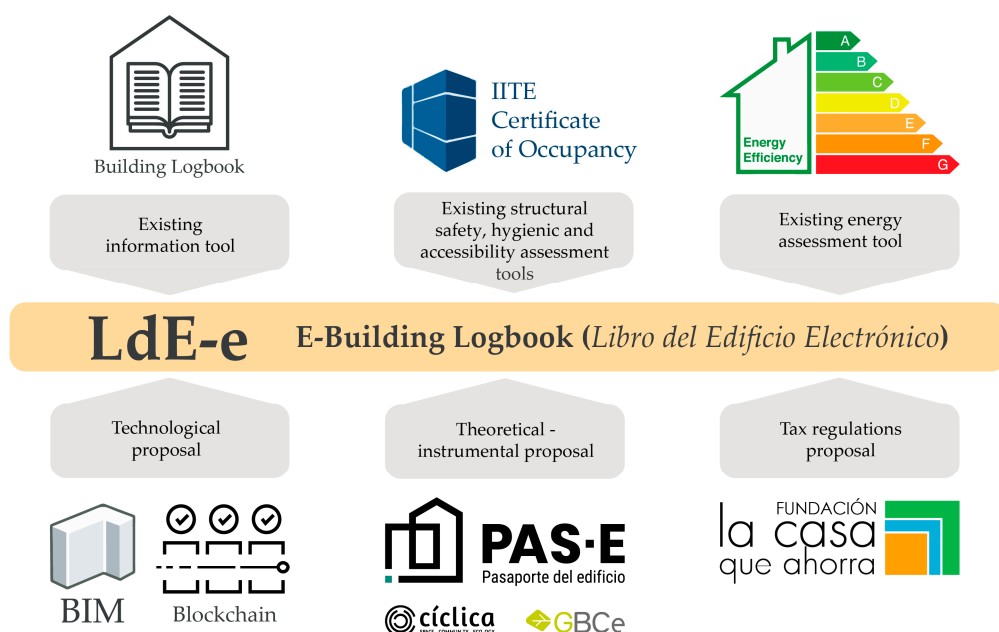

**Figure 2.** Initial elements for the LdE-e proposal. Source: own elaboration.

### 3.1.1. Input from Building Information and Management Tools

In Spain, information on multifamily residential buildings is managed in isolation. Hence, there is a varied, heterogeneous set of instruments that diverge in their operation, depending on when their elaboration became prescriptive, as explained below.

- Building Logbook (LdE, the Spanish acronym of *Libro del Edificio*). The LdE is a mandatory instrument for new buildings (since 2000 at national level, although in other autonomous communities such as Catalonia it has been mandatory since 1993). It collects all the information generated during construction work. In addition, it includes a conservation manual that allows implementation of a maintenance plan for the building, in theory. The LdE is delivered to the owner(s) in printed format, and any intervention on the building is supposed to be added to the record. However, this tool is outdated in practice, especially in the case of multifamily buildings without a professional facility manager;

- Existing Building Logbook (LEDEX, the Spanish acronym of *Libro del Edificio Existente*). LEDEX is a recently created instrument, based on the regulations of rehabilitation aid programs (Next Generation EU). In other autonomous communities such as Catalonia, it has been requested since 2015 for existing buildings after extensive renovation [29]. It includes an assessment of a building's current condition, its potential for improvement, a use/maintenance manual, and an action plan for renovation [30]. LEDEX is designed to be the logbook for buildings constructed before the year 2000, so its data structure is quite extensive, to record as much information as possible about the building and the dwellings in it. However, due to the difficulty in physically identifying all the components that make up an existing building (there is no obligation to carry out a test to identify materials, insulation, etc.), the success of the data provision relies on the expertise of the technician. In any case, LEDEX is fed by both technical inspections and EPC that have been carried out beforehand. Like the LdE, the LEDEX is delivered to the owner(s) in printed format, and the owner(s) is responsible for attaching any subsequent records;

- Certificate of Occupancy (CdH, the Spanish acronym of *Cédula de Habitabilidad*). The CdH was created in 1937 [31] as a state mechanism to control and guarantee minimum hygienic conditions of dwellings, due to the deplorable hygienic–sanitary conditions of some dwellings at that time (Spanish Civil War). Since 1944, the minimum conditions that are regulated have been the minimum dimensions of housing units, the height,

the area of lighting openings and the ventilation of the dwelling [32]. Currently, a CdH is requested for first occupancy of houses (new construction) and for second and subsequent occupancies. CdH are issued after a visit by a competent technician and are necessary to contract utilities such as electricity or gas supply. Finally, regulation of CdH is within the authority of autonomous communities, so the minimum requirements are not standard throughout Spain. Each autonomous community establishes specific requirements [33];

- Energy Performance Certificate (EPC). The EPC was designed by the EPBD in 2002 (transposed in Spain in 2007 for new buildings, and in 2013 for existing buildings). Its goal is to bring transparency to buyers/tenants/sellers regarding energy consumption and $CO_2$ emissions of dwellings through a simple indicator [34]. The EPBD is based on the hypothesis that if users of future buildings are informed about the benefits of EE, then they will make decisions that favor the most efficient buildings. The effectiveness of this premise in Spain is controversial. Spanish stakeholders believe that energy information transparency has not arouse the interest of the demand and supply of energy-efficient buildings [13];

- The Technical Building Inspection Report (IITE, the Spanish acronym of *Informe de Inspección Técnica del Edificio*). The IITE was created in 2011 as part of the adoption of Euro Plus Pact measures, through which the Spanish government sought to ensure that foreclosures are carried out without giving rise to abusive situations or misappropriation of the affected assets [35] and as a reactivation measure for the crisis the country was going through at that time. The IITE is mandatory for buildings over 50 years old and includes an assessment of the state of conservation of the building and the assessment of basic accessibility conditions to give a rating (favorable/unfavorable) on the final state of the building with a Certificate of Aptitude [36]. In the case of incidents, the technician makes recommendations to solve the deficiencies. While the IITE assesses the conventional conditions of the building, it also integrates the EPC rating and the recommendations for improvement of EE in its register.

An analysis of the data structure of the existing instruments (the matrix of each instrument has been provided in the Data Availability Statement) showed that, as they operate separately, many data items are repeated or discarded and regenerated (even those that do not change over time, such as the location of the building, year of construction or historical inspection record), which confirms the existence of information gaps [18].

### 3.1.2. Theoretical Inputs from Other Spanish Logbook Proposals

The PAS-E proposal [37] was used as inspiration for the existing instruments to create the LdE-e (the authors used 3 of the 5 instruments analyzed in the previous subsection). PAS-E also includes an interactive cloud-based tool, which was used in our proposal to generate a single online database for all building information (conventional and energy-related). Likewise, the Community Counseling Plan (technical support to guide owners on building renovation possibilities and cost-effective scenarios) proposed in PAS-E was taken as a reference and reinterpreted as a one-stop-shop (OSS) in the LdE-e model, as an integrated management entity to promote the energy renovation of dwellings [38].

### 3.1.3. Regulatory Input from Other National Proposals

One of the biggest challenges for the development and implementation of an LdE-e in Spain is the wide range of fiscal measures that should boost the renovation of the housing stock. The proposal of Fundación La Casa que Ahorra [39] moves in this direction. It identifies possible actions (subsidies, bonuses, financing, fee reductions, etc.) to encourage homeowners to make EE improvements and to highlight the relevance of a tool such as LdE-e for managing building information and scheduling renovations. This field of study is constantly evolving. In Spain, there are currently reductions in personal income tax (IRPF) or reductions in real estate tax (IBI) for EE improvement works in residential buildings [40].

### 3.1.4. Inputs from New Information and Communication Technologies (ICT)

The EPBD promotes the use of ICT, digitization and building automation to support energy information. A novel tool such as LdE-e may well incorporate new technologies that help build a robust data exchange system. In construction, the use of working ecosystems through building information modeling (BIM) methodology and the transparency, immutability and consensus properties of blockchain technology are maximizing the collective benefit of various stakeholders through a collaborative philosophy. In this respect, the LdE-e proposal integrates BIM and blockchain from the following perspectives of use.

- BIM methodology. Theoretical contributions on increasing the value and integrity of energy information in buildings suggest the use of BIM methodology to better represent energy consumption, future energy planning and quantification of building renovation rates [41,42]. BIM could help to mitigate the energy performance gap (understood as the disparity between the energy consumption predicted at the design stage of buildings and the consumption during operation), since the effective energy performance could be calculated directly from the information related to BIM materials and components [43]. The integration of BIM in the LdE-e model is proposed through the incorporation of a file with IFC, a BIM data interoperability and exchange format that has already been adapted to Spanish national regulations through the eCOB standard [44];
- Blockchain technology. The rationale for using blockchain in a BRP addresses the problem of a possible lack of trust between stakeholders and achieving an honest system without the involvement of a direct intermediary [45]. Recent studies have argued the importance of the use of blockchain in the development of a BRP due to the dynamic behavior of information throughout the life cycle of buildings and the need for rigorous updating processes (blockchain-based smart contracts could be useful for implementing mandatory improvements resulting from IITE with deficiencies rated as "serious" or "very serious") [46].

From this document base, the first draft of the LdE-e tool was created for validation through interviews with the panel of experts.

### 3.2. Semi-Structured Expert Panel Interviews

Once the initial outline of the LdE-e had been completed, it was reformulated and validated through semi-structured interviews with specialists and experts in the construction, architecture, property development and marketing sectors, and with officers from regional and local government with jurisdiction in the field. The knowledge gathered in the interviews allowed us to refine the structure of the LdE-e proposal and to advance in the determination of other aspects such as governance, management, usability, data flow, stakeholders, and the impact/barriers of the inclusion of new technologies (BIM and blockchain). In Spain, it was crucial to consider experts' opinions, to study the perceived confidence in other instruments such as the EPC scheme [13].

In the interview process, it was important to obtain the opinions of informants from the government and from professional associations involved in the regulation and issuance of existing information/assessment instruments, and of the technical professionals who produce them.

### 3.2.1. Identification of the Panel of Experts

The panel of experts was made up of energy efficiency specialists, real estate developers/builders, property agents, officers from technician associations (property managers, surveyors and architects), and representatives of regional and local managers of technical inspections and energy efficiency (Table 1). In Spain, architects are trained and have professional competencies in technical fields which in other countries fall within the role of installation and structural engineers. At this point, we had the collaboration of the Director of the Energy Efficiency Area of the Catalan Energy Institute (ICAEN), who rec-

ommended experts in the field of energy efficiency and also provided feedback on the opinions expressed in the interviews.

**Table 1.** Panel of experts interviewed.

| Industry | Institution—Organization—Company | Expert | Position | Background Code |
|---|---|---|---|---|
| Technical professionals | Cíclica | Joaquim Arcas Abella | Co-Founder | TE |
| Technical professionals | Self-employed | Ana Albiol | - | TE |
| Technical professionals | Self-employed | Rubén Argudo Salguero | - | TE |
| Technical professionals | Self-employed | Montse Barcones Campo | - | TE |
| Technicians' association | Association of Quantity Surveyors, Technical Architects and Building Engineers of Barcelona (CAATEB) | Jordi Marrot i Ticó | Technical director | CP |
| Technicians' associations | Architecture and Sustainability (AuS) | Albert Cuchí Burgos | President | CP |
| Real estate professionals | Association of Developers and Constructors of Catalonia (APCE) | Marc Torrent Dedeu | General manager | PI |
| Real estate professionals | Barcelona Association of Real Estate Agents (API) | Anna Puigdevall Sagrera | General manager | PI |
| Real estate professionals | Barcelona Association of Real Estate Agents (API) | Gustavo López Pecho | Operations manager | PI |
| Real estate professionals | Association of Property Managers of Barcelona-Lleida (CAFBL) | Lorenzo Viñas Periz | General manager | PI |
| Real estate professionals | Association of Property Managers of Barcelona-Lleida (CAFBL) | Carlos Pérez | Advisor | PI |
| Government | Housing Agency of Catalonia | Eva Paris Sánchez | Area manager | AD |
| Government | Local Energy Agency of Barcelona, Barcelona City Council | Cristina Castells Guiu | General manager | AD |

To differentiate the profiles of the interviewees in the analysis of the responses, the following codes were used:

- Technical professionals (TE);
- Technicians' associations (CP);
- Property professional (PI);
- Regional and local government (AD);

3.2.2. Approach to and Undertaking of Interviews

The purpose of the interviews was to consider the interviewees' experience to determine the potential benefits and possible difficulties in implementing the LdE-e, and the actions required for its correct application.

Due to the interviewees' profiles, three interview guides were defined for the following: (a) government and professional associations; (b) property professionals; (c) technical professionals. In all cases, regardless of the specific features, each interview guide was organized into the following blocks:

- Unified registry of existing building information management tools;
- Individual registry of data for each dwelling;
- LdE-e for new multifamily housing;
- LdE-e governance;
- LdE-e management;
- Usefulness/barriers of BIM and blockchain;
- Role of OSS in fostering LdE-e built-in SRR;
- Public policies and data use;
- Barriers to adopting LdE-e;
- Requirement and cost of the LdE-e.

For each interview with an expert, a personalized script was prepared depending on their background. However, there was some flexibility to develop certain topics in greater depth or identify others that had not been foreseen. For the convenience of the interviewees, it was clarified that no textual quotations would be made that could lead to identification of the participant. Therefore, opinions were referenced to the expert's background code.

All interviews were conducted virtually and recorded with the permission of the expert. A complete text of the responses was extracted from each of the recordings and arranged according to the proposed thematic blocks. Then, the responses were interpreted using the hermeneutic method, navigating between the parts and the entire text to achieve a holistic understanding of the experts' opinions.

### 3.2.3. Analysis of the Interviews and Validation of the LdE-e Proposal

Based on the interview results, a matrix was created in which coinciding and divergent ideas were grouped to create a comprehensive account of the evaluation of the LdE-e proposal. The following aspects (Table 2) stand out from the analysis of the interviews.

**Table 2.** Analysis of interviews by blocks. Source: own elaboration.

| Analysis Blocks | Expert Panel's Opinions |
| --- | --- |
| Unified registry of existing building information management tools | In general, there was consensus matching the initial hypothesis of this paper on the unification of the existing technical documents (CdH, EPC; IITE), since the scope they have separately is not the same (TE). From the real estate perspective (PI), experts indicated the usefulness of a unified registry to consider all the possible incidents that have affected a building before a property transaction is completed. In addition, the need to digitalize the LdE-e and to interactively introduce and retrieve information was highlighted (CP). From the government (AD) side, LEDEX already operates as a unified registry EPC+IITE, but it is not a digital tool, which prevents dynamic data flows. Finally, several experts (AD, CP) pointed out that it is crucial to analyze the quality and accuracy of the information to be integrated in the LdE-e registry, especially if the tool belongs to the owners, who usually do not have technical knowledge. |
| Individual registry of data for each dwelling | Experts agreed on the importance of having complete information on a building and on each of the dwellings in it. The value of the current LdE was recognized; since its prescription, all buildings have an active management tool (AD). This should be extended to existing buildings with LEDEX. However, the lack of rigor in data entry was noted, and the fact that the existing tools are only designed to comply with administrative files (CP). |
| LdE-e for new multifamily housing | Since the LdE-e proposal presented to the panel is itself a novel approach, there was little feedback from the expert panel. However, the experts (CP) stressed the need for a single data structure that can be interoperable and shared with stakeholders who are specialized in sustainable building. In other words, the LdE-e should be a tool that can provide data for housing policy making, not just a data repository. |
| LdE-e governance | Regardless of the experts' background, the governance of the LdE-e was perceived as complex, especially given the fact that there are many stakeholders involved. Some experts suggested a more technical vision of the LdE-e, in which the tool should be hosted on a public portal and preferably governed by a specific entity. In addition, it was suggested that each of the registry's modules or data packages should be developed by specific technical associations based on their technical knowledge (CP). |
| LdE-e management | In this section, there were divergent opinions. Some experts argued that, due to the complexity of the processes and data, a tool such as LdE-e cannot be managed by a board of neighbors or property communities. Instead, it should be managed by a professional such as a property manager (TE). This opinion was accepted by other experts, who emphasized that the property manager is already the recipient of the current LdE when multifamily owners delegate this function to the manger. They argued that this situation should be maintained in the case of a digital tool such as LdE-e (CP). Even so, the figure of the "primary technician" became known. In Spain, technical associations promote the role of "primary technician": a professional who supports households and property managers in the maintenance and retrofit of buildings. Such a stakeholder could also manage a building's LdE-e (AD). However, other experts agree that the government should participate in the management of LdE-e. The tool should be deposited in a public server, as the government is responsible for granting permits to access the information (TE) (for example, the administration could grant permits to a construction company to consult the SRR of a given building when it makes a bid to the owners). Furthermore, it is argued that the information contained in LdE-e should not be managed by private third parties, even though it is recognized that public management has serious constrictions related to the division of competences at different government levels and in different departments that would hinder tool management (CP). Distrust about the possible illegal sale of data was also discussed. Finally, it was suggested that for the management of the LdE-e to be efficient and its use effective, the procedures for its use should be regulated and mandatory. Therefore, discretional adoption of the LdE-e would jeopardize its diffusion and usefulness (CP). |

**Table 2.** *Cont.*

| Analysis Blocks | Expert Panel's Opinions |
|---|---|
| Usefulness/barriers of BIM and blockchain | Notwithstanding the lack of complete agreement of experts, there was a widespread opinion that technologies such as BIM and blockchain could be of great use due to their organizational capabilities and data exploitation possibilities. In the case of BIM, some experts argued that a digital twin could provide a vision of a building's or a dwelling's hidden defects. It could help the property community to agree and commit to the scheduling of energy retrofits (TE, CP). In the public sphere, blockchain is adopted with caution. While it is seen as a promising technology, it has been argued that it should not be considered from a paternalistic perspective, since technicians are responsible for the veracity/accuracy of the information they deliver, and blockchain will not exercise control over this responsibility (AD). Similarly, it was emphasized that citizens have the right to know the accurate state of conservation of their building, and that is why the use of this technology is positively valued (AD). However, there were doubts about the difficulty of implementing blockchain in LdE-e, since the technological training of the technicians entails logistical and economic efforts that could delay the implementation of the LdE-e (CP), especially since simpler tools already exist that make data immutable, such as electronic signatures on PDF documents. It was recognized that blockchain could help to better track the life cycle of buildings. |
| Role of OSS in fostering LdE-e built-in SRR | Several experts (TE) stressed that the development of IITE has served as an opportunity to approach property communities, fostering the emergence of the figure of "primary technician", which has facilitated the monitoring of rehabilitation tasks and given them control over the retrofit roadmap. Other experts (TE) consider that the recently created Technical Retrofit Offices (OTR) can help property communities to identify the retrofit roadmap (i.e., SRR) embedded in LdE-e. These OTR could serve as a starting point for owners to obtain information, but then the action must be transferred from the office to the building manager (e.g., the "primary technician" or property manager). Several experts agreed that two figures could appear: the "primary technician" who accompanies technical decisions, and the social agent, who accompanies the phases of agreement and compromise between owners (TE). Opinions were divided on the role of OSS. Some experts suggested that all architects involved in building renovation already do the job of encouraging users to renovate, and that the implementation of OSS may be good but not effective, because owners will always choose the cheapest option (TE). Other experts were clearly in favor of the implementation of OSS (AD), since these entities inform homeowners about existing EE subsidies linked to Next Generation EU funding. |
| Public policies and data use | In this regard, the experts agreed on two basic subjects. The first was that it was positive to have a data set on the housing stock available to the government to support public policies (AD). The second, in a negative sense, was that some experts have doubts about the publicity of open data. While it was indicated that control of the data should be public (CP, TE, AD, PI), care should be taken with confidential and personal information, since it should not be focused on making profit. Therefore, a code of data use should be included in the LdE-e design. |
| Barriers to adopting LdE-e | All the experts (TE, AD, CP, PI) argued that energy efficiency is not a determining factor in the decision to buy or rent a house, or even to undertake a renovation process. They recognized that the situation has improved compared to three or five years ago, but that we are at a point where energy-retrofit awareness is really needed. From the government side, (AD) indicated that the increase in the price of electricity or gas could be an opportunity to raise awareness and inform people about the potential of tools such as the LdE-e. Similarly, other experts (CP) emphasized that the information campaigns that have been carried out on Next Generation EU subsidies and grants are beginning to change people's mentality. However, beyond raising awareness, the upfront cost of energy retrofits remains a persistent barrier, and neighborhoods with low economic capacity will not be able to undertake renovation plans in the absence of comprehensive subsidies (PI). |
| Requirement and cost of LdE-e | In general, the experts did not have a clear view of the theoretical cost of developing the LdE-e. Some experts (TE) referred to existing BRPs in the European Union, but the cost approximations are not comparable because the economies are different from Spain. Other interviewees (CP) ventured to suggest that the possible cost of developing the LdE-e could be comparable to the development of a LEED, BREEAM or similar certification, but they were not sufficiently convinced. |

The identification of the role of each expert was crucial to give context to their opinions. For example, technical experts delved into specialized knowledge of building management, while regional and local government officials were more interested in the usefulness of the tool for the implementation of public policies. This divergence allowed for a holistic validation of the LdE-e proposal, which went beyond the strictly technical and extended

the study of the tool to vectors such as governance, management and the achievement of the EPBD guidelines.

## 4. Results and Discussion

Based on feedback from the expert panel, the structure of the proposed LdE-e was determined by reinterpreting the data in existing building information tools (Section 3.1.1), adapting the theoretical–instrumental and normative proposals put forward by Spanish agencies (Sections 3.1.2 and 3.1.3) and considering methodologies/technologies (Section 3.1.4) that can facilitate access to information by homeowners and stakeholders.

Therefore, the proposed LdE-e has a data structure that will be fed with information from a series of existing documents (documentary repository) throughout a building's life, at different stages of the life cycle. For this reason, the proposed data structure has been divided according to the condition of the building: new or existing.

### 4.1. Structure and Data Content of the LdE-e

In the specialized literature, the BRP is essentially composed of the DBL + SRR. The LdE-e model proposes a more comprehensive structure based on the adaptation of the information/assessment tools that exist in Spain and national proposals that were analyzed (Section 3.1).

The general structure of the LdE-e model, both for new construction (Figure 3) and existing buildings (Figure 4), consists of three main sections:

1.  Document Repository (DR), with the data assets of the existing documents and a supporting file package called Warehouse;
2.  Interconnected Data Structure (IDS);
3.  Scheduled Renovations Roadmap (SRR) throughout the life cycle of the building.

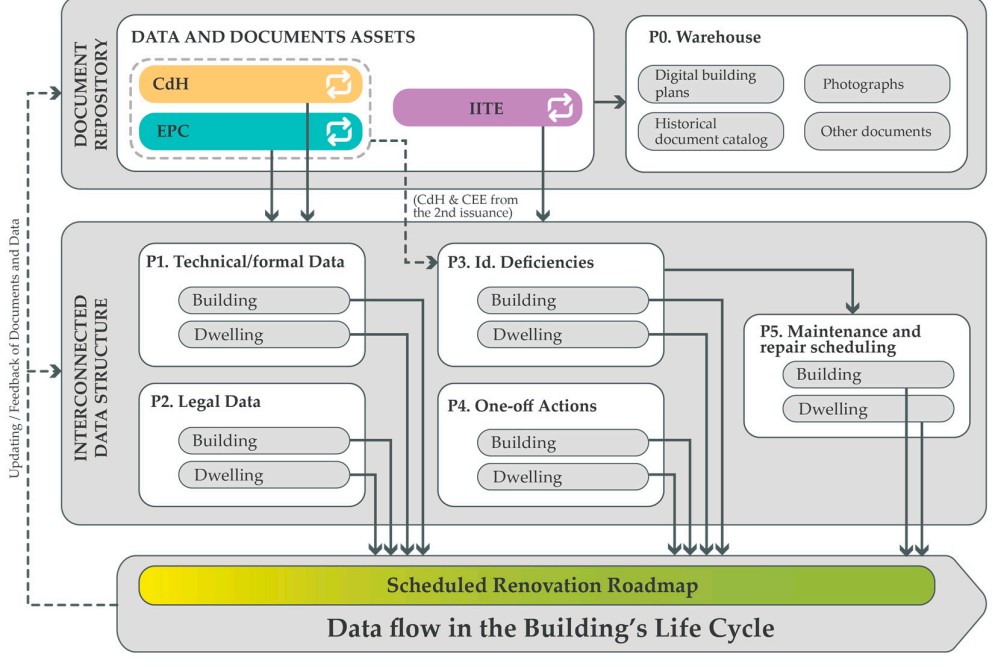

**Figure 3.** LdE-e for new buildings. Source: own elaboration.

**Figure 4.** LdE-e for existing buildings. Source: own elaboration.

### 4.1.1. Document Repository (DR)

The DR consist of two parts:

- Data and document assets. This contains the certificates resulting from the habitability (CdH), state of conservation (IITE) and energy efficiency (EPC) assessments. The information fields of these documents are read from the IDS information the first time they are issued. When such assessments are updated, and the resulting certificates reissued, data is retrieved from the IDS;

- Warehouse. This contains all the files that support the data (digital building plans, construction documents, building permits, deeds, warranties, policies, etc.). These documents will be stored in the warehouse as many times as they are issued for one-off actions or renovations throughout the building's life cycle.

### 4.1.2. Interconnected Data Structure (IDS)

This is a database structured in 5 packages that interact with each other. Each package contains a particular subject of data:

- P1. Technical/formal data. This contains the standardized description of the building's construction systems, the products, equipment and systems it contains (specific for each dwelling) and specific data to calculate the energy performance;

- P2. Legal data. This contains data on the agents involved throughout the building's life cycle, data on ownership (building and dwellings), and data on potential information requesters (as part of control on the provision of data to stakeholders);

- P3. Identification of deficiencies. This contains data fields referring to deficiencies in the structure, envelope, installations, accessibility conditions and the state of conservation of the building identified in the IITE. These fields will be empty until it is legally required to carry out the IITE;

- P4. One-off actions. This contains information on actions that do not derive from the IITE or the energy-retrofit recommendations contained in the EPC, but are incidents identified by the owners/users of the dwellings or their advisors (such as the "primary technician"), which are usually repaired without the need for a construction project requiring a building permit;
- P5. Maintenance and repair scheduling. This contains the record of scheduled maintenance operations and mandatory overhauls, operations to repair construction defects or flaws, common equipment, installations, etc. The actions derived from the SRR network are also recorded in this package. Similarly, this package contains the measures prescribed by the EPC to improve the terms of consumption, $CO_2$ emissions, heating and cooling demand, among others.

In the described data structure, some data will be introduced each time construction pathologies are identified, or when it is necessary to adapt the building and its dwellings to new regulations on accessibility, habitability or environmental sustainability. This is the main difference between the LdE-e of new or existing constructions, because a new building has a CdH and EPC from the start of its operation (Figure 3), but it will not have an IITE until it is 50 years old (according to current legislation). In any case, both the CdH and the EPC must be renewed after 25 and 10 years respectively, and from their second issuance they will also provide data to identify building deficiencies (P3).

In the case of existing buildings (Figure 4), the origin of the data is different. The inspections/certificates that give rise to the data structure are the EPC and the IITE for each dwelling that makes up the building (this implies an evolution of the LEDEX, which only computes building data). Eventually, if a second-occupancy CdH is issued (if the dwelling is sold, rented or modified), this document will also be stored in the DR.

Notably, the LdE-e proposal suggests the inclusion of information at building scale and for each of the dwellings, in response to the convenience of recording the state of the building's common elements (structure, exterior walls, roof, installations, common areas, etc.) and its private areas (dwellings). It contains complete information on the building's performance and state of conservation and the need for maintenance, reform or rehabilitation.

### 4.1.3. Scheduled Renovation Roadmap (SRR)

The IDS facilitates the elaboration of the SRR which, as it is connected to the DR, can feed back the renewal of the certificates (CdH, EPC and IITE) when necessary or mandatory. With the interconnection of data, it will be possible to refine the prioritization of renovations according to technical, economic and financial criteria, the functional needs of the owners/users, and their aesthetic aspirations. The operation of the SRR is detailed in Section 4.5.

### 4.2. LdE-e Data Flow

Although the structure and proposed content of the two LdE-e (new and existing buildings) are similar, the flow of information is not. Therefore, the analysis of data flows is considered according to the data direction, that is, whether the data are obtained reading (R) previously recorded data or data supplied (S) after inspection/issuance of a certificate. The data flow analysis for the LdE-e for new buildings (Table 3) and for existing buildings (Table 4) is presented below. Each package is disaggregated according to the sub-packages it contains.

Notably, the data flow for new buildings LdE-e (Table 3) is bidirectional in most cases. This is due to the gradual elaboration (Supply flow, S) of the instruments (first the CdH and the EPC, then the IITE) and to the expiry restrictions of the instruments. They must be renewed each time they become outdated, and previous data must be reprinted (Reading flow, R). The study of data flows in the LdE-e is relevant in terms of the stringency of data entry. Technicians may be more rigorous when they prepare reports, since the information may be useful to them in the future.

**Table 3.** Data flow for new building LdE-e. Source: own elaboration.

| Package | Sub-Package | CdH | EPC | IITE |
|---|---|---|---|---|
| P1. Technical/formal data | Building identification | - | R*/S | R/S |
| | General building data | - | - | R/S |
| | Description of the building | - | - | R/S |
| | Identification of dwellings | R*/S | R | - |
| | Standardized description of the building's construction systems | - | R*/S | R/S |
| | Description of products, equipment and systems | - | R*/S | R/S |
| | Energy performance data (building/dwelling) | - | R*/S | R |
| P2. Legal data | Data on the agents involved in the building's life cycle | R*/S | R*/S | R/S |
| | Property data | R/S | R/S | R |
| | Data on information requesters | S | S | S |
| | Data on the validity of the documents | S | S | S |
| P3. ID deficiencies | List and rating of the deficiencies found in inspections | - | - | R*/S |
| P4. One-off actions | Incident data in the building's life cycle | - | - | - |
| P5. Maintenance and repair scheduling | Record of maintenance and repair operations | - | - | - |
| | Technical recommendations for improving sustainability | - | R*/S | R*/S |
| P0. Warehouse | Supporting documents | R | R/S | R/S |

* Since the second issuance of the document.

**Table 4.** Data flow for existing building LdE-e. Source: own elaboration.

| Package | Sub-Package | IITE | EPC | CdH |
|---|---|---|---|---|
| P1. Technical/formal data | Building identification | S | R*/S | - |
| | General building data | R*/S | - | - |
| | Description of the building | R*/s | - | - |
| | Identification of dwellings | - | R*/S | R*/S |
| | Standardized description of the building's construction systems | R*/S | - | - |
| | Description of products, equipment and systems | R*/S | R*/S | - |
| | Energy performance data (building/dwelling) | R | R*/S | - |
| P2. Legal data | Data on the agents involved in the building's life cycle | S | S | R/S |
| | Property data | S | S | R |
| | Data on information requesters | S | S | S |
| | Data on the validity of the documents | S | S | S |
| P3. ID deficiencies | List and rating of the deficiencies found in inspections | R*/S | - | - |
| P4. One-off actions | Incident data in the building's life cycle | - | - | - |
| P5. Maintenance and repair scheduling | Record of maintenance and repair operations | - | - | - |
| | Technical recommendations for improving sustainability | R*/S | R*/S | - |
| | Safety of use and accessibility according to the Spanish Construction Code (CTE) | - | - | - |
| | Fire safety according to CTE | - | - | - |
| | Safety of the building according to CTE | - | - | - |
| | Protection against breakage according to CTE | - | - | - |
| | Action plan for the building renovation | - | - | - |
| P0. Warehouse | Supporting documents | R/S | R/S | R |

* Since the second issuance of the document.

In the case of existing buildings and the LdE-e (Table 4), we start from the premise that, as they existed prior to the year 2000, there is no physical documentary repository of information on the building. Therefore, the basic documentary instrument in this case is the IITE, which is the first-hand supplier of technical/formal data (P1), legal data (P2) and the identification of deficiencies (P3). However, when a new IITE is issued, the technician retrieves (reads) the previous data record. The pre-existence of the building

also extends the data record in the P5 package of the existing building's LdE-e, since the technical characteristics of the existing building must be evaluated according to the minimum requirements introduced by the current Spanish Construction Code (CTE) (RD 314/2006). Therefore, it is compulsory to assess the fulfillment of such requirements in the field of safety, fire protection, noise insulation, energy demand and accessibility.

The inclusion of these data records in P5 is suggested based on the harmonization of criteria of the proposed LdE-e for existing buildings with the LEDEX introduced by RD 853/2021 [30]. In this sense, our LdE-e proposal incorporates the content of the LEDEX evaluations and proposes the unification of the interconnected data registry.

### 4.3. LdE-e Governance at Institutional Level

As evidenced in the analysis of the expert panel interviews (Section 3.2.3), regardless of their background, the governance of the LdE-e is perceived as complex due to the large number of stakeholders that would be involved in the development and maintenance of the scheme. In the case of buildings' energy retrofits in Spain, researchers have highlighted the important vertical (i.e., levels of government) and horizontal (i.e., departments within each level of government) fragmentation that hinders the transition to decarbonized building stock [47]. This administrative fragmentation is also perceived as a barrier to the governance of the LdE-e by a single administration.

One possible form of organization to govern the operation of LdE-e is that of a consortium. In Spain, public law entities have their own distinct legal personality and are made up of several public administrations or institutional public sector entities with the participation of private entities. Governance by means of network integration is also guaranteed in the consortium figure. In all cases, in the constitution of one of these entities, the principle of neutrality applies whereby no agent can be privileged because of its affiliation or legal nature.

By creating a LdE-e consortium, government fragmentation may be reduced. The design and operation of LdE-e could be promoted by forming a new entity composed of representatives of different government scales/departments and representatives of technicians, properties, users, developers, constructors, suppliers and facility/property managers. Furthermore, formation of a consortium would bring advantages in terms of promoting technical or scientific cooperation for better, faster adoption of the LdE-e scheme. In the same way, the proposal for the foundation of an LdE-e Consortium is compatible with the need to address the retrofitting challenge from an integrated perspective, which has been identified as one of the main structural challenges of the Long-term Strategy for Energy Retrofitting in the Spanish Building Sector (ERESEE) in its latest version of 2020 [27].

### 4.4. LdE-e Management at Building Level

This proposal suggests a multi-scale level (Figure 5) for individual management of each LdE-e.

- As usual, the data and documents contained in the LdE-e would belong to the building/dwelling owners;
- Specialized data entry from inspections, projects and technical building interventions would be carried out by the respective technicians;
- Primary technicians, as recognized by the Long-term Strategy for Energy Rehabilitation in the Building Sector in Spain (ERESEE, 2020), and property managers would oversee the correct updating and maintenance of LdE-e information throughout the building's life cycle;
- The LdE-e Consortium would have custody of the instrument, in addition to competences regarding the maintenance of the IT platform, and the governance of the LdE-e. The LdE-e Consortium would be the body through which third parties interested in building information could access aggregated data, excluding protected or sensitive personal data.

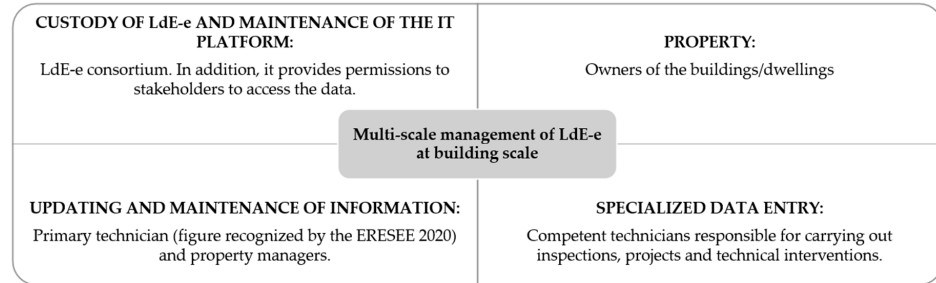

**Figure 5.** Multi-scale management LdE-e. Source: own elaboration.

*4.5. Scheduled Renovation Roadmap*

The SRR is the third major component of the LdE-e structure for existing buildings and new ones (when it becomes necessary). It is an action plan that is fed by all the interconnected data sets. Eventually, it will serve as feedback or to update the document repository and the interconnected data sets, due to the foreseeable physical and technical improvement of the intervening buildings.

Figure 6 shows the structure of the implementation of an SRR. It indicates the four main stakeholders involved and their roles (explained in detail below):

1.  Owners and users;
2.  The OSS;
3.  The primary technician;
4.  The installer or construction company;

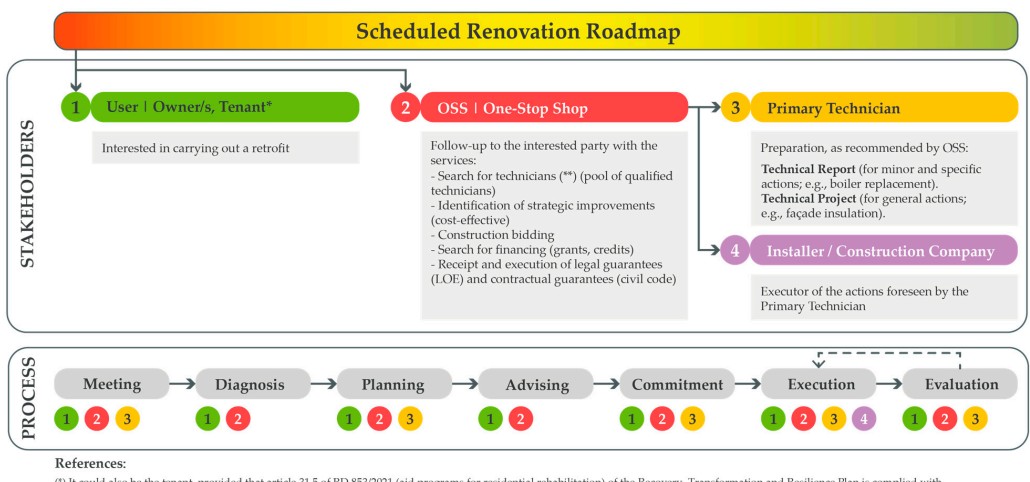

**Figure 6.** SRR stakeholders and process. Source: own elaboration.

These four stakeholders will be present in some of the phases of the process, depending on their function within a scheme of seven steps:

1.  Meeting;
2.  Diagnosis;
3.  Planning;
4.  Advising;
5.  Commitment;
6.  Execution;
7.  Evaluation (of the execution of the previous phase).

Since the activation of the SRR is still voluntary, the first and main agent will always be the owner (or property communities). If the owner is not aware of the benefits of energy renovation, combined with other possible improvements, or does not find the situation plausible and reliable to initiate a renovation process, he/she will simply not

activate any renovation program. Hence the relevance of the second stakeholder: the OSS. The intervention of the OSS is proposed as a strategy for democratic access to support and to provide competent advice on energy rehabilitation. This figure complements the important task that some technicians and property managers already do in terms of advice, support and prescription of building improvements. In fact, the first OSS that has started to operate in Spain has strong links with the associations of property managers, since these professionals are the gateway to homeowners' associations and play the role of mediators [48].

Process for Scheduling Renovations

After the implementation of the OSS, the figures of primary technician and installer or construction company are deployed. This proposal of organizational dependence is important, since the embryonic figure of the OSS in Spain is, above all, informative [48]. The OSS proposal presented here takes a step forward and proposes comprehensively delivering the required services throughout the process of drafting and executing the SRR.

The SRR tailoring process consists of seven steps through which consensus is reached among the four intervening agents. Table 5 provides a detailed description of each of the steps in the process.

**Table 5.** SRR tailoring process. Source: own elaboration.

| N° | Process Stage | Agents Involved | Description of Stage |
|---|---|---|---|
| 1 | Meeting | Owner OSS Primary technician | The owners (*) initiate (**) the process by communicating with OSS to develop the SRR. In this first phase, the figure of the primary technician is defined (chosen by the owners, from a pool of qualified technicians), who will oversee the preparation of the technical report or the technical project according to the technical assistance of the OSS. A technician chosen by the OSS will carry out the inspection visit to the building and will rely on the information provided in the IITE or the EPC. |
| 2 | Diagnosis | Owner OSS | Based on the inspection and technical assistance of the OSS, a qualitative diagnostic meeting is held, in which the OSS informs the owners of the condition of the building and whether specific or general adaptations are advisable. In addition to the technical criteria, the OSS should consider the functional needs and aesthetic aspirations of the owners. |
| 3 | Planning | Owner OSS Primary technician | Based on the OSS diagnosis, the primary technician recommends and proposes a report or project of actions to the owners, with its planning by stages and the cost-effective analysis. The primary technician can present several planning proposals, so that the owners can choose the option that best suits their possibilities. The data required for this recommendation could be retrieved from the LdE-e data fields or, in case of first issue data, supplied to the IDS packages. |
| 4 | Advising | Owner OSS | Once the draft/proposed stages of the actions have been defined, the OSS discusses the investment figures with the owners, and advises on the possibilities of financing, aid or credits that could be useful, explaining their financial advantages and disadvantages. |
| 5 | Commitment | Owner OSS Primary technician | Once the owners have been technically and financially advised, a commitment to implement renovations (including energy retrofits) in stages is agreed. The expected financial and non-financial benefits, the estimated costs of each renovation stage, and the impact on the building's energy rating are indicated. In this regard, the primary technician must draw up a technical report or project for each renovation phase, based on the recommendations (technical assistance) made by the OSS. |
| 6 | Execution | Owner OSS Primary technician Installer/Construction company | The OSS provides a labor pool of installers/construction companies that execute the first stage of renovations programmed according to the primary technician's proposal, with the prior consent of the owners. |
| 7 | Evaluation | Owner OSS Primary technician | The OSS, with the help of the primary technician, supervises the execution and identifies the degree of implementation of the prescribed actions. If the actions are modified, the primary technician will have to reformulate the next stage of renovation with the consent of the owners. If the action has followed the specifications of the report or project, the owners are informed of the programming of the next stage of execution. Each time a step is completed, the primary technician must record new data (e.g., improvements made and impact on the building's energy rating) in the IDS. This evaluation process will be iterative until all the planned renovation stages have been completed. In any case, new intervention needs may arise because of new IITE, new EPC or changes in building regulations. |

\* By owners we mean the property community. \*\* Although in the seven-step scheme we indicate that the owner initiates the process, OSS usually have a "stage 0" that contributes to the achievement of the process by disseminating the benefits and co-benefits of energy renovation. In addition, at this "stage 0" a first contact is established with the property community to inform them about the advantages of the renovations and to make a first evaluation of the need for intervention in the building. Failing this, the property manager also acts as a liaison between the owners and the OSS.

As can be seen, the participation of the owner is required throughout the process, since it is the agent that initiates the SRR (currently, on a voluntary basis). Without the owner's decision, the SRR is simply not activated.

In this regard, an important factor is the approval of Law 10/2022, on urgent measures to promote building rehabilitation activity [49], which amended Law 49/1960 on Horizontal Property [28]. Since this amendment, rehabilitation works that contribute to the improvement of a building's energy efficiency may be carried out with the favorable vote of 3/5 of the owners. Dissenting neighbors must also cover the loan or financing granted, as the costs are considered general expenses for the building's maintenance. In any case, all owners must understand the financial and non-financial benefits of energy efficiency to avoid tensions among them when agreeing to SRR financing.

### 4.6. BIM in the LdE-e

The approach of integrating BIM into LdE-e is limited to new buildings, given the ease of using the methodology from the initial process of building design and construction. Information contained in the digital twin would make it possible to determine in detail the material constitution of the building in all stages and to calculate energy and emissions performance. In the case of existing buildings, it is unfeasible to request the provision of a digital twin, given the technical difficulties of acquiring precise materials and non-recordable elements (such as internal insulation of walls, installations integrated into the structure, etc.) that make up the building, including original materials and those used in past retrofits.

Although BIM methodology can be implemented for design and analysis of the energy performance of existing building renovations, the scale of the intervention may be an obstacle for technicians to consider its use. This would not be a problem in the case of buildings which, in their new construction condition, included the BIM files in the LdE-e's Warehouse (P0). In this case, for subsequent renovation proposals, the technician would have the BIM file and could use it as a basis for proposing renovations.

To implement BIM in the LdE-e, the use of the data exchange format called IFC is proposed. The relevance of the IFC as a building data exchange format is given by its "tree structure". This structure is like the one we use in the LdE-e proposal, in which "root" packages are structured and from which sub-packages with interconnected building data fields are derived (Figure 7).

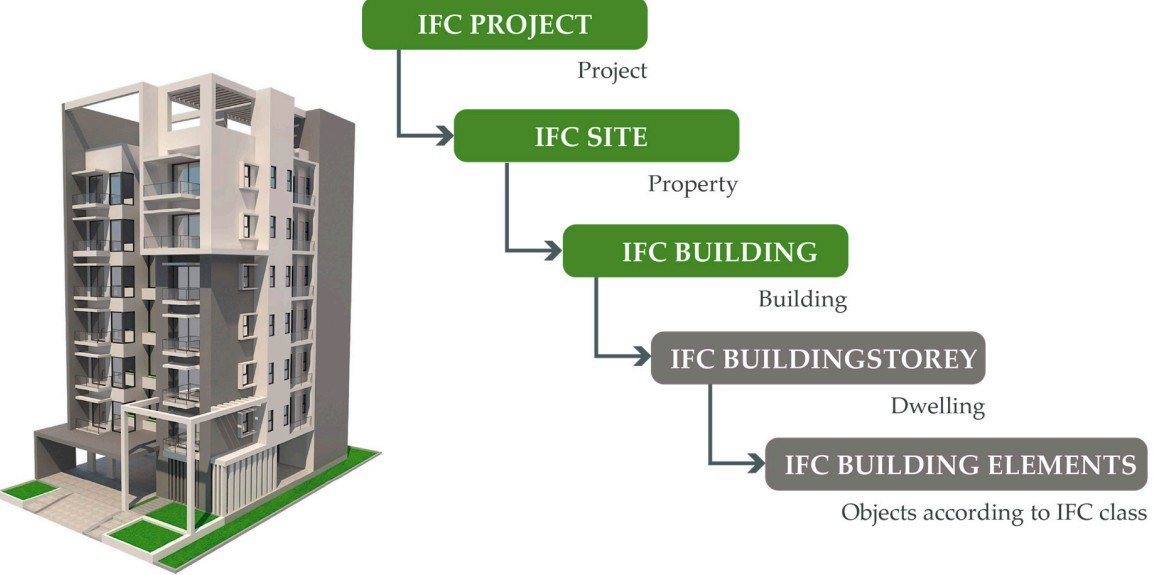

**Figure 7.** Tree structure of the IFC format. Source: own elaboration based on www.buildbim.cl (accessed on 16 December 2022).

The use of the IFC scheme when a new building is designed makes it possible to define the building elements, prefabricated products, mechanical systems, electrical safety systems or generalized analyses of the building such as their mechanical-structural behavior or energy performance, etc. Through the standardization of data semantics, this information could easily be supplied in the data fields defined for the LdE-e (Figure 8).

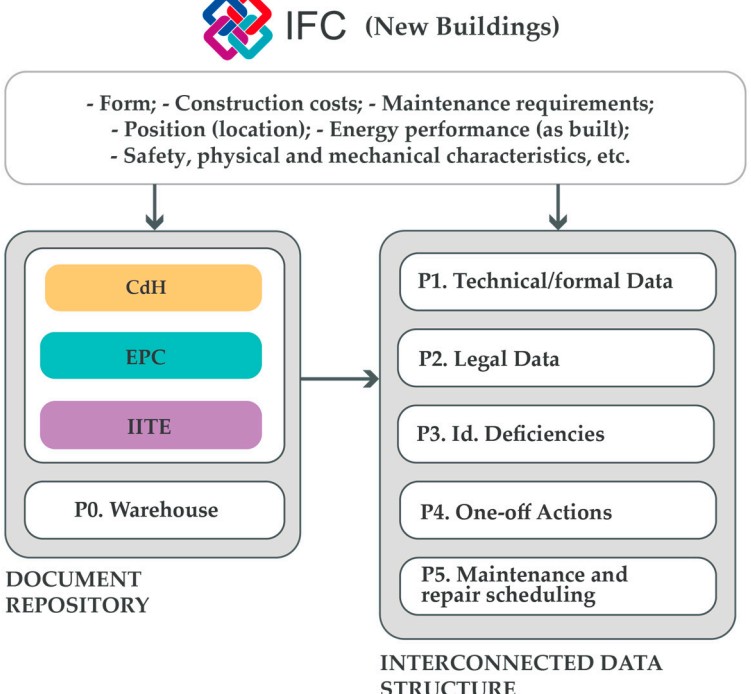

**Figure 8.** Implementation of the IFC (BIM) format in the new building LdE-e. Source: own elaboration.

Thus, by integrating the building IFC file in LdE-e (as built), all the particularities of its design and construction (identified even for each dwelling) would be included, and the technicians could retrieve this data for the elaboration of technical inspections (CdH, EPC, IITE), and supply or modify data in the IFC file when the actions or renovations of the building change the configuration and technical values.

### 4.7. Blockchain in the LdE-e

The preliminary approach to the interaction of the blockchain with the LdE-e was the conformation of a complex ecosystem in which all the data records (or transactions to the blockchain semantics) would be consigned in a permissioned blockchain. However, the results of the expert panel interviews (Section 3.2.3) showed that there were generally some reservations about the technological approach of a blockchain-based LdE-e. The experts' reluctance was justified by the arduous task of training technicians in the technology supporting the tool, or by the existence of simpler ways of securing building data (such as electronic signatures in PDF files).

Therefore, the degree of implementation of blockchain in this LdE-e proposal has been reformulated to a less complex degree, which could be scaled to incorporate a more sophisticated ecosystem in the future. To this end, the proposed implementation of blockchain in the LdE-e is based on the tokenization of assets, understood as the final certificates of each of the instruments that make up the documentary repository: (a) CdH; (b) EPC (not the label, but the certificate); (c) IITE (not the Certificate of Aptitude, but the preliminary report).

Possibly, the LdE-e Consortium could evaluate the need to tokenize technical intervention projects, executive projects for building extensions or even renovation projects.

The proposal for tokenization of assets (certificates and eventually projects) is shown graphically in Figure 9. It is proposed that the agents in charge of tokenizing the certificates

(and eventually projects) are the same technicians who perform the inspections/assessments that originate them, since they will have a better opportunity to perform the procedure as part of the assignment requested by the property community. The tokenization of these certificates makes sense because they are documents that contain sensitive information on the building, to which third parties may need access to check the record (credentials) with the certainty that the data contained are reliable, authentic and reflect the reality of the real estate.

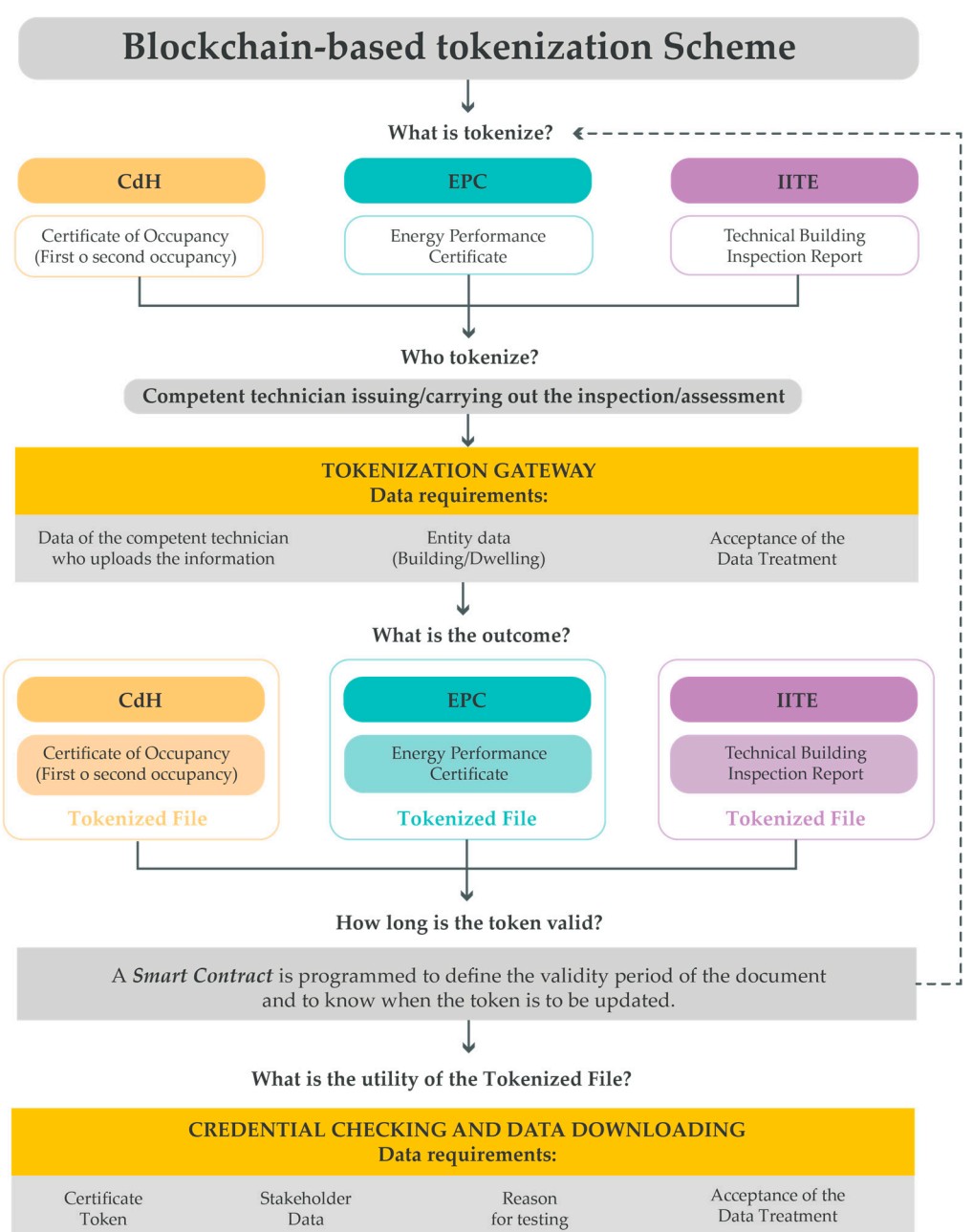

**Figure 9.** Blockchain-based tokenization scheme. Source: own elaboration.

The information required to initiate the certificate tokenization process could be:
- Data of the technician who uploads the information;
- Entity data (cadastral reference of the house or building, if applicable);
- Acceptance of the data processing.

Based on these requirements, access is needed to a tokenization gateway that returns the tokenized certificates. However, all certificates have a temporary validity, which is why we propose the use of a Smart Contract (second generation blockchain technology utility) to indicate the period of validity of the token to the owner and third parties accessing the content.

The complexity of using blockchain in the proposed LdE-e could be scaled depending on future requirements. A future implementation utility could be stakeholder monitoring or validation of data read permissions for interested third parties.

*4.8. Advantages of the Future Implementation of the LdE-e*

The main advantage of the proposed LdE-e is that it would contain the "biography" of the building in a single digital repository, with a set of information fields that would feed, in part, the evaluations performed in residential buildings throughout their life (such as the CdH, EPC and IITE). At the same time, these evaluations carried out through on-site inspections would feed new fields (e.g., those related to the energy efficiency of the building or those related to the necessary renovations), which would make it possible to have a reliable "snapshot" of the state of the buildings at all times.

Thus, the LdE-e would improve, among other aspects:

- Management of the building, for example, by controlling its operation and maintenance more easily;
- Evaluation of the building, for example, facilitating the work of the technicians who carry out mandatory inspections;
- Time programming of improvements based on identification of the deficiencies in the technical inspections, and the needs, aspirations and possibilities of the owners and users with the financial assistance of public and private systems;
- Improvement planning, from the drafting of the renovation and one-off intervention projects;
- The cost of the improvement and the market value increase derived from the execution.

In addition, having an updated photograph of the buildings would allow, on an aggregate scale:

- An improvement in the diagnosis of residential stock with a view to drawing up public policies to improve the quality of housing in all its dimensions (with the appropriate restrictions, while the rental housing stock plays a marginal role in housing renovation decisions, due to trigger points between the owner and the tenant);
- An improvement in the databases focused on real estate taxation and, therefore, on having a more solid base to finance public and social policies;
- Quantification of business opportunities for the manufacture of materials and equipment, their distribution, the provision of construction services, development, installation, replacement of equipment and materials, and building maintenance;
- Diagnosis of the state of the residential stock with a view to the activities of the insurance and financial sector, and in this way a reduction in the risk implicit in the private financing of building rehabilitation actions.

All the above applications are in line with Article 14 of the current EPBD recast draft, which refers to the exchange of data, in terms of access to data by owners, lessees, managers and interested third parties.

Finally, an LdE-e based on information and communication neo-technologies such as blockchain and BIM would allow:

- Improvement in the authenticity of information about the building, and on the documents resulting from its evaluation and design through the inspection of the traceability of its authors/issuers;
- Improvement in the authenticity of the information would allow a more accurate evaluation of the implicit risk in purchase and rental transactions and in financing, including an increase in the building's performance, especially in terms of energy;

- Enablement of the use of blockchain technology, such as smart contracts, which would allow automation of the validity periods of tokenized certificates for the owner and stakeholders interested in the content;
- Information to be obtained on the technical performance, economic and environmental implications of each building's materials and construction systems. This information will become more relevant, as shown by the EPBD recast draft requiring new buildings to include greenhouse gas emissions, implicit in the entire life cycle of the built-in elements, from the extraction of raw materials, their transformation, transport, maintenance and eventual dismantling. With this information, the global warming potential (GWP) could be calculated in an expeditious manner and made known to current and prospective users of buildings through the recast EPC.

The interviews with thirteen experts led us to the conclusion that it is imperative to integrate the BRP into the framework of a much more ambitious instrument. The opinions of the panel of experts were essential to compare the starting hypotheses of this work. The experts' answers have had a significant impact on the proposed LdE-e. The theoretical conception of this LdE-e proposal is a genuine breakthrough not only for Spain, but also as a cutting-edge approach to the development and adoption of these tools in the European Union.

### 4.9. Limitations

Due to the context of the proposed LdE-e model, the results of this study are limited to the Spanish context. However, the choice of the panel of experts was based on the professional relationships and local knowledge of representatives of institutions interested in sustainable building stock. Therefore, it might be necessary to extend the vision of the tool to national representatives of institutions such as the Institute for Energy Diversification and Saving (IDAE) or the Ministry of Transport, Mobility and Urban Agenda (MITMA). In addition, as future lines of research are necessary to include the integration of financial data packages, it would be interesting to extend the validation of the LdE-e model to financial agents with expertise in sustainable building (i.e., green finance).

### 4.10. Future Research Lines

The development of an LdE-e requires the involvement of stakeholders from different areas of expertise. The proposal presented in this document is a simple step forward in the construction of the structure, management, governance and technological implementation of the LdE-e, which can be used as a reference for aspects that require further research, such as:

- The cost of developing and implementing an LdE-e, its financing phases and the economic means to achieve it;
- The regulatory barriers that discourage the renovation of residential buildings, and whose effects would detract from the usefulness of the LdE-e;
- The changes in the legislation that regulates construction to enable the existence of the LdE-e;
- The organic structure that the LdE-e Consortium should have and its governance to combine and reconcile the interests of private agents, civil society and government levels and departments with authority in building, housing, energy efficiency and other authorities that could improve their public function because of better knowledge of the built-up area;
- The study of BIM implementation policies in Spain while its use in private residential real estate developments is still voluntary. This would encourage the use of the methodology in the LdE-e;
- The study of the integration of singular data packages in the LdE-e, such as a financial analysis of real estate, an evaluation of the increase in value of the buildings due to the execution of improvements or their comparative maintenance or renovation costs,

which would be valuable information for the owner, for the public administration, and for third parties interested in the sustainable real estate market.

## 5. Conclusions

The main contribution of this study is to establish a preliminary from basis to advance towards an E-Building Logbook (LdE-e). This cloud-based tool would contain all the information related to the life cycle of a building, i.e., when it is designed, built and used. After some time, the building is evaluated for its habitability, safety, health, accessibility and energy efficiency. On this basis (which would make up what is known as DBL), combined with functional and aesthetic aspirations and financial and management possibilities, an SRR is proposed, which is executed and put into operation until the building is dismantled.

The proposed LdE-e is fully consistent with some of the aspects raised by the European Commission in the current EPBD recast. Thus, for example, the LdE-e could serve as the basic tool for the diagnosis, by means of an EPC, of future compliance with the MEPS. This would account for the "scaling" of energy ratings of buildings which, according to Article 9 of the EPBD recast draft, Spain will have to implement by 2030 in the case of residential buildings. At the same time, the LdE-e proposal is in line with the spirit of the Digital Building Register of the EPBD. This registry is a common repository of all building data, including related data on energy efficiency, scheduled renovations and readiness indicators for smart applications.

Likewise, by 2030, the EPBD draft introduces an indicator called global warming potential (GWP), which is calculated according to the Level(s) framework, to report on emissions over the full life cycle of new buildings. With the implementation of BIM in the LdE-e and the collection of materiality data (form, structure and materials) through the IFC format, the future calculation of the GWP indicator could be easily incorporated (as an additional data field in the physical and technical data package, P1) when indicated by the EPBD.

In view of the opinions of the expert panel, the benefits of the future application of the LdE-e are essential to overcome the informational barriers that slow down the decarbonization of the Spanish housing stock. The challenge is enormous, and it remains to put into practice a prototype LdE-e, to determine, in particular, its effectiveness in terms of use by homeowners. Notably, in view of the voluntary nature of the renovation process, the owner has the first and last word.

**Author Contributions:** Conceptualization, P.E.-Z. and C.M.-D., revised by A.G.-H.; methodology, P.E.-Z. and C.M.-D.; validation, P.E.-Z., C.M.-D. and A.G.-H.; formal analysis, P.E.-Z. and C.M.-D.; investigation, P.E.-Z., C.M.-D. and A.G.-H.; data curation, P.E.-Z.; writing—original draft preparation, P.E.-Z.; writing—review and editing, C.M.-D. and A.G.-H.; visualization, P.E.-Z.; supervision, C.M.-D.; project administration, C.M.-D.; funding acquisition, C.M.-D. All authors have read and agreed to the published version of the manuscript.

**Funding:** This study was carried out as part of the doctoral thesis of the first author framed in the project Beyond the EPC Requirements, potential and risks of energy efficient mortgages in the promotion of efficient homes, EnerValor2, funded by the Spanish Ministry of Science, Innovation and Universities (ref. PID2019-104561RB-I00). Similarly, some of the results were obtained due to a grant for research work on public administration and public policy from the Escola d'Administració Pública de Catalunya (Public Administration School of Catalonia) (ref. BDNS 545679).

**Institutional Review Board Statement:** Not applicable.

**Informed Consent Statement:** Informed consent was obtained from all subjects involved in the study.

**Data Availability Statement:** The complete matrix of each of the data fields that make up the Interconnected Data Packages can be found in the following link: https://drive.google.com/drive/folders/1w0LMgt4faTudlWS2QTszFA-vBq40EcTg?usp=sharing (accessed on 27 December 2022).

**Acknowledgments:** The authors greatly appreciate the detailed contributions and criticism of reviewers. The authors would also like to thank Lluís Morer, Head of the Energy Saving and Efficiency

Department of the Catalan Energy Institute (ICAEN) and Rolando Biere, ETSAB assistant professor and CPSV researcher, for their valuable contribution to the production of this document. The authors are grateful to the experts who were interviewed and are solely responsible for the interpretation of the interviewees' opinions.

**Conflicts of Interest:** The authors declare no conflict of interest.

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
