# Peer review of "Libro del Edificio Electrónico (LdE-e): Advancing towards a Comprehensive Tool for the Management and Renovation of Multifamily Buildings in Spain"

_sustainability, doi:10.3390/su15042957_

Round 1

Reviewer 1 Report

The paper presents an interesting topic and it is ready to be published; however, minor changes are required before it’s publication:

Move figure 2 after the first paragraph of section 4.

Line 200 - what do you mean with the volumen of lighting openings? The area of the openings for lighting and ventilation? Line 427 - the word appear isn’t necessary. Table 4 - under which instrument is the evaluation of the minimum requirements of the CTE (P5) performed? Line 542 - change is for are. Table 5 - please describe better the process since the role of owners (mostly related to decision making) is overseen. Additionally, here or in other part of the text it should be discussed the complexity of this type of decisions in multi-familiar buildings since a minimum of property owners have to agree to perform such renovations. Line 575 - eliminate LdE-e at the end of the sentence. Line 584-585 - reformulate this sentence. It cannot be understood. Line 589 - at him/her disposal.

Reviewer 2 Report

Overall interestign study. Below points can be improved: 

Figure 1, text size and style shall follow templates. Color of the boxes can be in calmer tones. ANd the size of the figure slightly smaller. 

Point 3. there is usually no questionmark in the tittle. 

Perhaps points: 2,3,4,5 shall be under Methodology. 

The methodology shall be re-organized. Add a paragraph explaining the main steps followed. 

Results are welldone. 

Reviewer 3 Report

Overall, it is a profound, well-written, and well-explained study. I have no major objection to highlighting. Only have two minor suggestions to the author, if he agrees with me:

·      The structure of the paper consists of too many sections (almost 8) and a few sections are very small. I would suggest the author to merge or put some sections under one heading. This will improve the outline and structure of the paper and will be easy for a reader to follow. For example, section 1 and section 2 can be merged. Section 3 and section 4 can be put under one main heading. Section 5 can be “Methodology for Interviews” or any suitable precise/ pertinent heading. Section 6 and section 7 can be placed under the heading “Results & Discussion”.  

·      The bullets/ improvement points mentioned in the Conclusion section shouldn’t be part of the Conclusion. Those can be shifted to the previous section, a separate section, or a sub-section. The Conclusion seems unnecessarily long.

Reviewer 4 Report

Please see my suggestions in  sustainability-2156063-peer-review-v1 copy attached 

Round 2

Reviewer 2 Report

Minor formating issues. 

Good improvement from the last manusctipt. 
